# Impact of COVID-19 on Healthcare Labor Market in the United States: Lower Paid Workers Experienced Higher Vulnerability and Slower Recovery

**DOI:** 10.3390/ijerph18083894

**Published:** 2021-04-08

**Authors:** Neeraj Bhandari, Kavita Batra, Soumya Upadhyay, Christopher Cochran

**Affiliations:** 1Department of Healthcare Administration and Policy, School of Public Health, University of Nevada, Las Vegas, NV 89119, USA; soumya.upadhyay@unlv.edu (S.U.); chris.cochran@unlv.edu (C.C.); 2Office of Research, School of Medicine, University of Nevada, Las Vegas, NV 89102, USA; Kavita.batra@unlv.edu

**Keywords:** COVID-19, healthcare employment, current population survey, labor market

## Abstract

The resilience of the healthcare industry, often considered recession-proof, is being tested by the COVID-19 induced reductions in physical mobility and restrictions on elective and non-emergent medical procedures. We assess early COVID-19 effects on the dynamics of decline and recovery in healthcare labor markets in the United States. Descriptive analyses with monthly cross-sectional data on unemployment rates, employment, labor market entry/exit, and weekly work hours among healthcare workers in each healthcare industry and occupation, using the Current Population Survey from July 2019−2020 were performed. We found that unemployment rates increased dramatically for all healthcare industries, with the strongest early impacts on dentists’ offices (41.3%), outpatient centers (10.5%), physician offices (9.5%), and home health (7.8%). Lower paid workers such as technologists/technicians (10.5%) and healthcare aides (12.6%) were hit hardest and faced persistently high unemployment, while nurses (4%), physicians/surgeons (1.4%), and pharmacists (0.7%) were spared major disruptions. Unique economic vulnerabilities faced by low-income healthcare workers may need to be addressed to avoid serious disruptions from future events similar to COVID-19.

## 1. Introduction

The COVID-19 pandemic has severely impacted international and US labor markets [1,2,3,4]. Healthcare jobs are often considered less vulnerable to typical economic recessions than other sectors of the economy. During the 2009 recession, while the overall economy lost 8 million jobs, the healthcare industry added 850,000 new jobs [5]. Many health problems are “demand inelastic”; people visit their doctors in good and bad times [6]. The resilience of the healthcare industry is being severely tested in the COVID-19 induced recession, which is unlike any other recession in the past. The pandemic led to sharp reductions in physical mobility along with institutional and legal restrictions on elective and non-emergent medical procedures, followed in short order by reports of massive layoffs, furloughs, and temporary job separations in the healthcare industry [7,8,9]. Elective procedures are financially critical for many hospitals since they underwrite operations that are more critical but less profitable [9]. As a result, strained ERs and critical care units in major urban centers are struggling to respond to successive waves of COVID-19 cases rippling across the nation [10]. Elective services form an even larger part of the business model of outpatient providers [11]. Additionally, many physicians in outpatient settings have turned towards telehealth options to serve patients, who could no longer be seen in person, further reducing the need for front-end office personnel. Surveys show that nearly half of the practices have furloughed employees, with many laying off their employees permanently [7].

In this paper, we assess the impacts of COVID-19 on employment and labor market dynamics of the US healthcare industry. Varying susceptibility of different types of healthcare jobs to the pandemic may generate differential impacts in terms of job losses and recovery across different sectors. Healthcare consumers predictably reduced the use of medical procedures that allowed prescheduling or for conditions that impose limited morbidity without medical care [12]. Early reports suggest that technicians and aides in the dentist and primary care offices were the earliest to be affected by job losses [13]. These disruptions caught many hospitals short-handed, when new waves of infections appeared in affected communities and may have prompted the hiring of more medical and support staff to accommodate higher demand for care [14,15]. However, care-seeking for non-COVID critical illness plummeted at the same time [16,17], causing critical headwinds that may have mitigated or even reduced hiring. Similarly, it is unclear what effect the pandemic has had on the jobs of healthcare workforce employed in long term care facilities and those delivering home health care. Some states instituted policies that directed a steady stream of hospital discharged COVID-19 inpatients discharged to skilled nursing facilities (SNFs), in part to limit community spread [18]. In the early phase of this pandemic, Centers for Medicare and Medicaid Services (CMS) released guidance for ensuring isolation of COVID-19 patients from other patients [19], which may have prompted the flow of resources towards costly workflow redesigns. Collectively, these policies could have spurred hiring. At the same time, states and the federal government suspended expensive regulatory compliance procedures and periodic inspections [20], granting room for relaxed workflow processes and easing pressures for hiring personnel. Using the latest Current Population Survey (CPS) data, we provide the first empirical evidence of early COVID-19 effects on multiple measures of health and resilience of labor markets for key sectors of the healthcare industry and occupations. Our study provides a clear snapshot of the COVID-19 induced dynamics of decline and recovery in healthcare job markets and may be helpful in informing policy responses to future rare events such as this pandemic, particularly its impact on low-wage healthcare workers such as healthcare aides, therapists, and technicians.

## 2. Materials and Methods

We performed a monthly trend analysis of key employment outcomes using data from the basic monthly public use files of the CPS data from July 2019–2020 [21]. The CPS collects nationally representative data on the non-institutionalized US civilian population, including detailed information on labor force participation, employment status, work hours, occupation, and industry. We focused our analyses on workers aged 16–70 years, who reported being in the healthcare labor force, after excluding those in active military duty and those who reported being retired or disabled (*n* = 418,899). The survey administrators used the 2012 North American Industry Classification System (NAICS) and the 2010 Standard Occupational Classification (SOC) systems to identify healthcare industries and occupations in 2019 but switched to 2017 NAICS and 2018 SOC starting January 2020 [22]. We regrouped detailed industry and occupation subcategories into several broader categories to facilitate interpretation. Our final industry categories are hospitals, home healthcare services, physician offices, dental offices, nursing care facilities, residential care facilities, and outpatient centers. Hospitals include general, medical, and specialty hospitals but exclude psychiatric and substance abuse hospitals. Home health care services include (1) home health care services, (2) health practitioners other than physicians, dentists, optometrists, and chiropractors, and (3) miscellaneous outpatient health care services that do not fit into any other category. Healthcare occupation groups include physicians and surgeons, dentists, healthcare aides, pharmacists, technicians, nurses, mid-level practitioners, and other therapists. Regroupings for healthcare occupations are as follows: (1)Physician group combines emergency medicine physicians, radiologists, anesthesiologists, cardiologists, dermatologists, family medicine physicians, general internal medicine physicians, neurologists, obstetricians and gynecologists, general pediatricians, pathologist physicians, psychiatrists, and a miscellaneous category.(2)Healthcare aides include home health aides, personal care aides, nursing assistants, orderlies and psychiatric aides, occupational therapy assistants and aides, physical therapist assistants and aides, massage therapists, dental assistants, medical assistants, medical transcriptionists, pharmacy aides, veterinary assistants and laboratory animal caretakers, phlebotomists, and miscellaneous healthcare support workers.(3)Technicians include dental hygienists, cardiovascular technologists and technicians, diagnostic medical sonographers, radiologic technologists and technicians, magnetic resonance imaging technologists, nuclear medicine technologists and medical dosimetrists, emergency medical technicians, paramedics, pharmacy technicians, psychiatric technicians, surgical technologists, veterinary technologists and technicians, dietetic technicians and ophthalmic medical technicians, medical records specialists, dispensing opticians, miscellaneous health technologists and technicians, and a miscellaneous category for other healthcare practitioners and technical occupations.(4)Nurses include registered nurses, licensed practical nurses, and licensed vocational nurses.(5)Midlevel practitioners include physician assistants, nurse anesthetists, nurse practitioners, and nurse midwives.(6)“Other therapists” include chiropractors, dietitians and nutritionists, optometrists, podiatrists, audiologists, occupational therapists, physical therapists, radiation therapists, recreational therapists, respiratory therapists, speech-language pathologists, acupuncturists, a miscellaneous category for all other therapists, and a miscellaneous category for all other healthcare diagnosing or treating practitioners.

We assigned respondents to these groupings based on their primary job status when they reported multiple jobs. For a few categories that saw minor changes due to changes in the coding classification, we used the crosslinking guide provided by the Census Bureau to maintain similarity between groupings across 2019 and 2020 [22]. For each industry and occupation, we generated aggregate monthly unemployment rates, month-to-month change in absolute volume of employment and labor market entry/exit, and percent change in aggregate weekly hours worked, using the composite employment weight provided by the Census Bureau. The unemployment rate was calculated as an industry-wise or occupation-wise share of labor force that reported being unemployed. Change in employment indicates the month-to-month volume of job losses/gains by type of industry or occupation and is estimated by the formula Ʃi wgti × Li, where Li equals 1 if person i is employed and 0 otherwise. The employment weight wgti represents person i’s share of total employment in the civilian labor force. Change in labor force participation signifies month-to-month change in volume of exit/entry into a specific industry or occupation, and is estimated by formula Ʃi wgti × Li, where Li equals 1 if person i is in the labor force (either employed or unemployed) and 0 otherwise. Finally, the percent change in total hours worked was computed as month-to-month change in a cumulative number of weekly hours worked by the type of industry and is estimated by formula Ʃi wgti × Li × Hi where Li equals 1 if person i is employed and 0 otherwise, and Hi indicates person i’s actual hours worked in the reference week. Stata version 16 (College Station, TX, USA: Stata Corp LP) was used for analyses. Further, since our data are not seasonally adjusted, we computed monthly trends for all employment outcomes for March to June 2019 as a baseline to compare COVID-related trends in March–June 2020.

## 3. Results

Figure 1 plots monthly trends in unemployment rates by the healthcare industry, color-coded to reflect the sharp transition between the pre-COVID-19 period (until February 2020) and the enactment of shelter-in-place orders and the concomitant sharp curtailment in economic activity (March 2020 onwards). The unemployment rate was the highest in dental offices, which rose from 4.3% in March 2020 to peak at 41.3% in April 2020, before steadily declining to 8.9% in July 2020. Notably, this was significantly higher than the overall national unemployment rate of 14.7% in April 2020 [23]. In outpatient centers, the unemployment rate increased from March 2020 to April 2020 (0.8% to 10.5%) and then significantly declined to 3.8% in July 2020. The unemployment rate for physician offices rose by 7.8 percentage points from 1.7% in December 2019 to peak at 9.5% in April 2020 (Figure 1). There was an initial drop in June to 5.7%, which reversed in July when the rate climbed back to 8.6%. 

The unemployment rate in residential care facilities grew more slowly, from 4.4% in March 2020 to peak at 8.6% in May, followed by a steady decline to 4.4% in July 2020. The increase in the unemployment rate in hospitals was notably muted compared to other sectors, increasing slightly from 1.6% in March to 4.4% by June 2020. The unemployment rate in home healthcare services rose from 2.5% in March to peak at 7.8% in April 2020, decreased slightly for two consecutive months before reaching 7.6% in July 2020. Compared with other healthcare industries, nursing care facilities had lesser disruption but more volatility in terms of the unemployment rate, which increased from 3.4% in March 2020 to peak at 6.9% in April 2020, and then declined in May (4.6%), followed by an increase to 6.5% in July 2020. Among the healthcare occupations, the unemployment rates increased in April 2020 for dentists (19.0%), healthcare aides (12.6%), technicians (10.5%), other therapists (8.9%), mid-level practitioners (7.1%), nurses (4.0%), and physician and surgeons (1.4%) (Figure 2). This measure peaked in May for dentists before steeply declining to zero percent in July 2020. Physicians and surgeons experienced a spike in the unemployment rate in July 2020, after a drop in June 2020. Appendix A, Table A1 and Table A2 provide additional information on trends in unemployment rates from March to June 2019. Comparing the same months in March–June 2020 reveals large increases that become evident around April 2020 and follow a consistent pattern across all industries and occupations, indicating that 2020 changes are not solely seasonal fluctuations.

COVID-19 seems to have dramatically accelerated the already high degree of churn in healthcare labor markets. During March–April 2020, 327,000 workers in home health services, 273,000 in hospitals, and 133,000 in dental offices exited the labor market (Figure 5). However, net labor force entry was observed in nursing care facilities (*n* = 70,000), residential care facilities (*n* = 39,000), and outpatient centers (*n* = 38,000) during the same period. Amongst occupations, again excepting mid-level practitioners, we see a high labor market exit (Figure 6). Most sectors also saw reductions in aggregate hours worked between March and April, with especially sharp declines recorded for dentists (−80%) and physician offices (−17.2%). The number of aggregate hours started recovering slowly for all industries (except residential care facilities) during April–May 2020 (Appendix A, Figure A1). The number of aggregate hours worked increased after April among all occupations (except for mid-level practitioners and pharmacists) (Appendix A, Figure A2). 

## 4. Discussion

Consistent with expectations, our findings suggest that the pandemic had disproportionately negative impacts on employment in healthcare industries and occupations that provided care for procedures that could be postponed. Our findings support anecdotal media reports of widespread closure of dental offices in the initial wave of job losses [24]. Employees working in dentist offices saw unemployment rates climb to unprecedented levels in March, exceeding 40%, with swift recovery in ensuing months. This may be partly because dental care is less amenable to provision via telehealth and dental practices face considerable barriers in adapting to virtual provision of services [25]. Similar but less intense job losses were seen for those working in outpatient care centers and physicians’ offices. 

Unemployment among workers at hospitals and nursing care facilities grew more modestly and slowly, even as both types of facilities saw substantial job losses in the early months of the pandemic. Indeed, unemployment rates across hospitals were increasing as late as June. COVID-19 caseloads may have spurred hiring in some regions, but the effects could have been offset by layoffs following a swift drop-off in elective procedures and care for critical non-COVID-19 illness. The fact that the total hours worked remained steady despite job losses suggests the remaining workforce worked more hours. This is consistent with reports of hospitals imposing extended work schedules to cope with the first wave of COVID-19-related hospitalizations affecting the Northeast region and later the Sunbelt south [25]. Interestingly, a significant labor market exit from hospitals started in February, just after the emergence of a cluster of COVID-19 cases in the northwest. It is possible that some workers left the hospital workforce fearing COVID-19 impacts on their personal safety, even before involuntary job losses due to slacking demand began to accelerate. Whatever the reason, this degree of turnover probably helped some sectors avoid recording even higher unemployment rates. Nursing homes initially saw small net job gains despite higher unemployment due to the increased labor market entry but had large job losses and significant labor market exit by June, indicating a delayed impact of the pandemic. Residential care facilities weathered the crisis better but typically offer lower paying jobs compared to nursing homes. Whether nursing homes shed more employees (relative to residential care) due to heavier COVID-19 mortality among their residents should be examined in future research. The severity of job losses at home health care agencies fell somewhere between those faced by outpatient care and hospitals, likely because mobility restrictions forced several scheduled home visits to be cancelled [26].

Among occupations, healthcare aides and technicians seem to have borne the brunt of job losses while nurses and pharmacists were spared major disruptions. This may partly reflect the skewed composition of the workforce in hospitals and physician offices in favor of positions filled with healthcare aides and technologists. It may also reflect the limited fungibility of skills needed to do these jobs, making efficient substitution less feasible and costlier. Nurses fill a unique niche in the hospital workforce and may seamlessly transfer job skills across cross-subsidized operations. In a similar vein, demand for many prescription drugs tends to be inelastic, and most pharmacies remained open during the crises, both factors providing pharmacists a significant cushion against a sudden collapse of revenue. 

The differential pattern of recovery also holds out lessons for policymakers. Dentists’ offices saw the steepest job losses but rebounded quickly, leading the initial recovery in the overall job market. Yet even here, unemployment rates remained in excess of 10% at the end of July 2020, suggesting some of the losses may be permanent. Employment in physician offices recovered modestly, climbing again in July, potentially indicating continued softness in demand for outpatient/primary care due to COVID-19. This contrasted with outpatient care centers, which saw a fuller recovery suggesting growing confidence among patients to pursue elective surgeries and lab procedures. Recovery among home healthcare workers, healthcare aides, and technologists was slow to begin with, and remained woefully incomplete with persistently high unemployment rates. 

Our results tentatively suggest that the pandemic’s burden fell more on lower-income employees and support the need for policy measures to bolster the job security of these employees. Many healthcare organizations remain exceedingly reliant on revenue streams from elective procedures, which put them at high risk of failure during a pandemic [7]. If pandemics remain once-in-century events, then fashioning a policy cushion against such losses may not make much practical sense so as to be cost-effective. However, the frequency of major epidemics has increased, possibly due to enhanced environmental pressures associated with continued human encroachment into virgin ecosystems [27]. Several policies may help mitigate economic impacts on vulnerable organizations and employees. Federal financial support for healthcare employers, automatically triggered by a formal declaration of a pandemic, may help sustain strategic furloughs and avoid permanent layoffs. For clinics, COVID-19 induced realignment in daily operations towards telehealth delivery since a broadening range of clinical services could help sustain revenues and avoid sharp layoffs. Federal and state funds should continue to support and subsidize employee training and other costs in starting and maintaining this shift. Large employers may be required to set aside a portion of their revenues to maintain rainy day funds that help them tide over drying revenue streams and avoid immediate disruptions. Other pandemic trigger options can include deregulation of policies that make sense during normal times but impose unnecessary hardships for employers trying to retain temporarily unproductive employees. Finally, existing payment reform initiatives that predate the pandemic could be tailored to reduce reliance on elective procedures, e.g., reducing the use of fee-for-service payment systems (that often overpay for high-profit technologically intensive elective services such as knee or hip joint implantation and underpay primary care services) in favor of bundled and capitation provider fees.

Several fertile areas of research are implied based on these results. The healthcare profession has long been considered as a recession-proof industry. For example, during the great recession downturn from 2007–2010, the US unemployment rate rose to 10%. However, there was little or no impact on the healthcare industry regardless of occupational setting or geographic location [5]. There were some indications of a slowdown in healthcare due to the effects of the Tax Cuts and Jobs Act of 2017, which eliminated tax penalties for uninsured individuals [28]. However, the data presented in this study indicates an accelerated slow-down over previous months. The healthcare industry employs a large number of non-clinical workers, particularly in outpatient settings. These workers typically earn less than clinicians. As state and local officials implemented lockdowns across the country, this left many workers struggling with decisions about caring for school age children as well as child day care centers. Evidence of these types of events exists for previous outbreaks that resulted in school closures for a much shorter duration [29]. Further research must examine whether these workers were forced to leave their jobs in order to be at home with their children. 

### Limitations 

Our study results need to be qualified with some limitations. First, COVID-19 depressed CPS household survey response rates significantly (by 10–15%), and most interviews were carried out on telephone rather than in-person, which generally yields more accurate data [30]. Second, the sheer volume of COVID-19 related job losses led to undercounting unemployment since a significant number of unemployed on temporary layoffs were misclassified as employed but absent from work [31]. Third, the reliability of employment estimates declines as the sample size shrinks, which may have introduced some measurement error for smaller industry and occupation groupings. To account for this, most of our groupings exceeded at least 500 respondents, with a few exceptions: Dentists’ offices, residential facilities, physicians/surgeons, dentists, and mid-level practitioners. Fourth, we neglected seasonality in hiring and firing since these adjustments are generally applied to a larger time series than ours. However, since seasonal fluctuations can be quite large in certain contexts [32], we provided a comparison between March–June 2019 and March–June 2020 trends to assuage concerns that 2020 changes may be merely seasonal fluctuations. Fifth, we used monthly BLS data from CPS to compute unemployment rather than the higher frequency and more granular information available through weekly state wise unemployment insurance claims. This limitation is mitigated by recent research suggesting that weekly claims through March–May 2020 may have substantially overestimated payroll losses during these initial months of pandemic and contributed to massive discrepancies between expectations and realized job losses [33]. Finally, our findings are not generalizable to healthcare systems outside the United States, which have significantly different governing structures and payment practices. For example, the private sector plays a disproportionately larger role in delivering care in the US compared to other OECD nations (e.g., United Kingdom, Canada), making the US healthcare labor force exceptionally vulnerable to market forces and economic downturns.

## 5. Conclusions

Healthcare jobs have often been viewed as recession-proof or largely immune from financial crises. Our findings suggest that low-income workers in healthcare support occupations face unique economic vulnerabilities during pandemic-induced financial crises. This may need to be addressed through a broad array of targeted reforms to avoid serious disruptions in both job security and productivity of the healthcare workforce resulting from future events similar to COVID-19. Future studies to assess the long-term impact of the COVID-19 pandemic and mitigation policies/directives targeting labor force inequities will be beneficial to ameliorate its long-term impacts.

## Figures and Tables

**Figure 1 ijerph-18-03894-f001:**
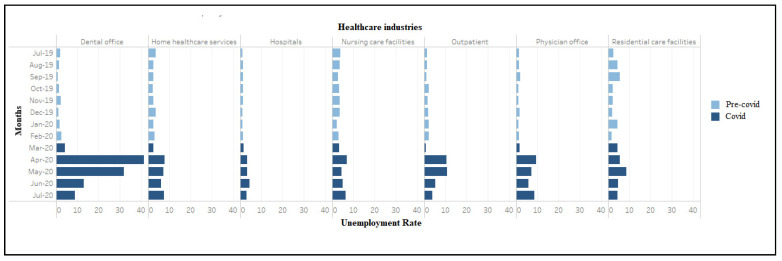
Unemployment rate (weighted) among workers in healthcare industries (July 2019–2020).

**Figure 2 ijerph-18-03894-f002:**
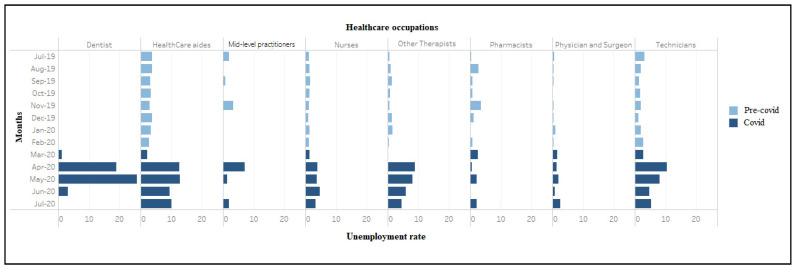
Unemployment rate (weighted) among workers in healthcare occupations (July 2019–2020). During March–April 2020, significant job losses occurred in home health services (*n* = 490,000), dental offices (*n* = 442,000), hospitals (*n* = 394,000), outpatient centers (*n* = 156,000), and physician offices (*n* = 152,000) (Figure 3). Conversely, nursing care facilities (*n* = 15,000) and residential care facilities (*n* = 25,000) showed small job gains. Large job losses also occurred in virtually every healthcare occupation between March–April 2020, including healthcare aides (*n* = 769,000), technicians (*n* = 339,000), therapists other than physicians and surgeons (*n* = 148,000), nurses (*n* = 122,000), physicians and surgeons (*n* = 63,000) and dentists (56,000), with an exception of 2000 jobs gained among mid-level practitioners (Figure 4).

**Figure 3 ijerph-18-03894-f003:**
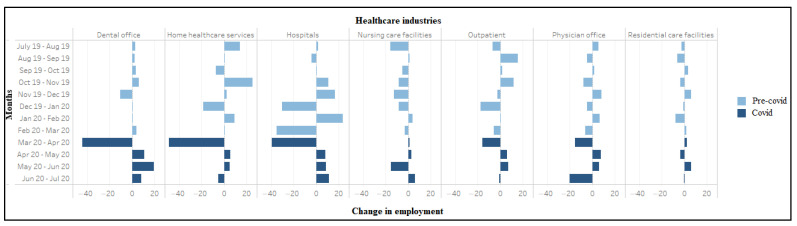
Change in employment (in ten thousand) among workers in healthcare industries (July 2019–2020).

**Figure 4 ijerph-18-03894-f004:**
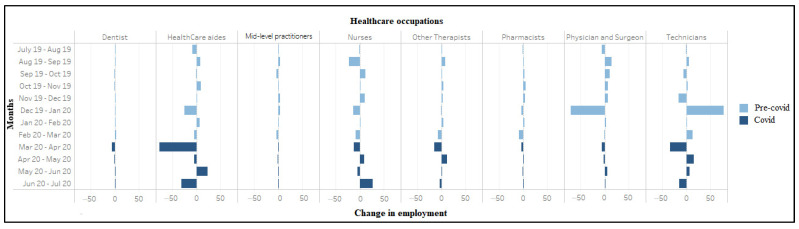
Change in employment (in ten thousand) among workers in healthcare occupations (July 2019–2020).

**Figure 5 ijerph-18-03894-f005:**
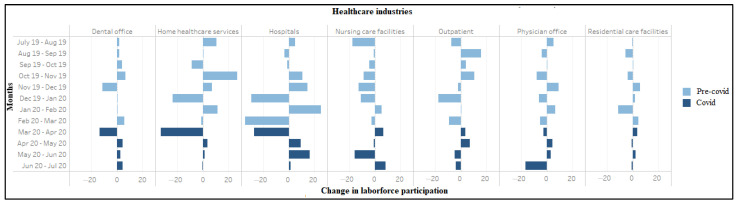
Change in labor force participation (in ten thousand) among workers in healthcare industries (July 2019–2020).

**Figure 6 ijerph-18-03894-f006:**
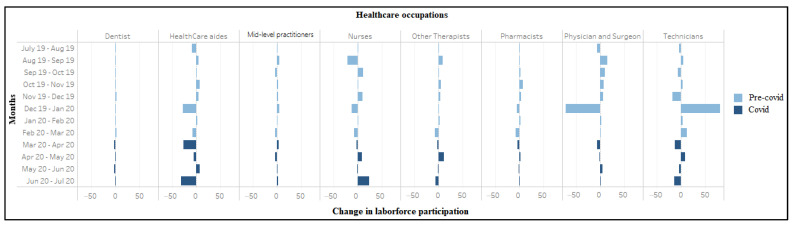
Change in labor force participation (in ten thousand) among workers in healthcare occupations (July 2019–2020).

## Data Availability

The datasets generated during and/or analyzed during the current study are available in the US Census Bureau repository: https://www.census.gov/data/datasets/time-series/demo/cps/cps-basic.html (accessed on: 15 January 2021).

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
