# Peer review of "Impact of COVID-19 on Healthcare Labor Market in the United States: Lower Paid Workers Experienced Higher Vulnerability and Slower Recovery"

_ijerph, 2021, doi:10.3390/ijerph18083894_

Round 1
Reviewer 1 Report
- The article is clear and well-written. The importance of the issue is explicated and relevant context data is references to support the up-to-date knowledge about the impact of the pandemics on the healthcare sector.
- Please, include the fact that this is US data in the abstract and title. This is relevant, because health economics and healthcare system differ across the globe. Discuss whether there are any limitations resulting from the way the system is designed in the US.
- Consider placing Figures using landscape orientation - the data is hard to read (even on the external monitor).
- Some data follows seasonality, and it is true for the unemployment in certain sectors. Typically, seasonally adjusted data is reported for unemployment rates to reveal the underlying trends and cycles in labor markets (Shiskin & Plewes, 1978). I believe it would be valuable if the Authors compared their "covid data" to the same months the previous year. Thus, March-July 2019 should be provided as a reference.
- Please, comment to what extent the unemployment benefits in the initial phases of the pandemic may have affected the data. Cajner et al. 2020 argue that insured unemployment offers important advantages as a barometer of labor market conditions. Discuss, whether this approach limits your findings and in what way.
References
Cajner, Tomaz, Andrew Figura, Brendan M. Price, David Ratner, and Alison Weingarden (2020). “Reconciling Unemployment Claims with Job Losses in the First Months of the COVID-19 Crisis,” Finance and Economics Discussion Series 2020-055. Washington: Board of Governors of the Federal Reserve System, https://doi.org/10.17016/FEDS.2020.055.
Shiskin, J., & Plewes, T. J. (1978). Seasonal adjustment of the US unemployment rate. Journal of the Royal Statistical Society. Series D (The Statistician), 27(3/4), 181-202.
Reviewer 2 Report
Dear Authors,
The topic corresponds with current reader's interest.
Hovewer, in my opinion, the paper should have more author contribution (for example statistical analysis).
You should define the purpose of the research. What were you looking for? Did you manage to confirm the results?
The materials and methods section is described too superficial. In line 74 you should provide the link to database, or determine the direction of searching for source data or provide leads. Generally, it would be better to construct the framework (or algorithm), which will present the paths and documents analized. Such a solution is very often practiced if the research is based on the database analysis.
The related work section should be improved. Please consider the following positions:
https://www.ncbi.nlm.nih.gov/pmc/articles/PMC7661819/
https://www.ncbi.nlm.nih.gov/pmc/articles/PMC7670225/
Radulescu, C.V.; Ladaru, G.-R.; Burlacu, S.; Constantin, F. et al. Impact of the
COVID-19 Pandemic on the Romanian Labor Market. Sustainability 2021, 13, 271. https://doi.org/10.3390/su13010271
Shuai, X., Chmura, C. & Stinchcomb, J. COVID-19, labor demand, and government responses: evidence from job posting data. Bus Econ 56, 29–42 (2021). https://doi.org/10.1057/s11369-020-00192-2 https://link.springer.com/article/10.1057/s11369-020-00192-2#citeas
Too much electronical sources have been used.
Are your research is up-to-date? May be some new policies already have been introduced?
The reference [7] - without access.
The manuscript requires minor technical changes, ex. references style.
The conclusion and introduction sections could be extended. The quality of pictures should be improved. May be it will be better to change the font color on black.
I suggest you to change the title on "Impact of COVID-19 on healthcare labor market in United States" (Your research concern only US data).
Please describe better the methodology. Is it possible to provide the link to concrete CPS data? ex. line 84, line 88 (not only to the start web page). You also should better describe the data transformation process (including regrouping - line 82) and the limitations of this regrouping.
And, finally, that are benefits of obtained results in scientific tearms. How we can use them for scientific purposes?
You haven't study the correlation, are you sure that the changes are totaly the results of the pandemy?
line 312 "No matter the cause, these results highlight the pandemic's inequitable burden on lower-income employees and support the need for policy measures to bolster the job security of these employees" - I think you research is not enough to make such a conclusion.
Moreover, the conclusion used in title "Lower paid workers experienced higher vulnerability and slower recovery" should be definitely proved before publishing it.
Round 2
Reviewer 2 Report
Dear Authors,
Equations view could be improved (line47-52),
The conclusion could be extended with future work description.
Best regards.
